# Amlexanox: Readthrough Induction and Nonsense-Mediated mRNA Decay Inhibition in a Charcot–Marie–Tooth Model of hiPSCs-Derived Neuronal Cells Harboring a Nonsense Mutation in *GDAP1* Gene

**DOI:** 10.3390/ph16071034

**Published:** 2023-07-21

**Authors:** Nesrine Benslimane, Federica Miressi, Camille Loret, Laurence Richard, Angélique Nizou, Ioanna Pyromali, Pierre-Antoine Faye, Frédéric Favreau, Fabrice Lejeune, Anne-Sophie Lia

**Affiliations:** 1NeurIT UR 20218, GEIST Institute, Faculté de Médecine de Limoges, University of Limoges, F-87000 Limoges, France; nesrine.benslimane@hotmail.com (N.B.); f.miressi@gmail.com (F.M.); camille.loret@unilim.fr (C.L.); laurence.richard@unilim.fr (L.R.); angelique.nizou@unilim.fr (A.N.); ioanna.pyromali@unilim.fr (I.P.); pierre-antoine.faye@unilim.fr (P.-A.F.); frederic.favreau@unilim.fr (F.F.); 2CHU Limoges, Service de Neurologie, F-87000 Limoges, France; 3Centre Hospitalier Universitaire (CHU) Limoges, Service de Biochimie et de Génétique Moléculaire, F-87000 Limoges, France; 4CNRS, Inserm, CHU Lille, UMR9020-U1277—CANTHER—Cancer Heterogeneity Plasticity and Resistance to Therapies, University of Lille, F-59000 Lille, France; fabrice.lejeune@inserm.fr; 5Centre Hospitalo-Universitaire (CHU) Limoges, UF de Bioinformatique, F-87000 Limoges, France

**Keywords:** GDAP1, CMT disease, nonsense mutation, nonsense-mediated decay (NMD), readthrough molecules, premature termination codon (PTC)

## Abstract

Nonsense mutations are involved in multiple peripheral neuropathies. These mutations induce the presence of a premature termination codon (PTC) at the mRNA level. As a result, a dysfunctional or truncated protein is synthesized, or even absent linked to nonsense-mediated mRNA degradation (NMD) system activation. Readthrough molecules or NMD inhibitors could be innovative therapies in these hereditary neuropathies, particularly molecules harboring the dual activity as amlexanox. Charcot–Marie–Tooth (CMT) is the most common inherited pathology of the peripheral nervous system, affecting 1 in 2500 people worldwide. Nonsense mutations in the *GDAP1* gene have been associated with a severe form of CMT, prompting us to investigate the effect of readthrough and NMD inhibitor molecules. Although not clearly defined, GDAP1 could be involved in mitochondrial functions, such as mitophagy. We focused on the homozygous c.581C>G (p.Ser194*) mutation inducing CMT2H using patient human induced pluripotent stem cell (hiPSC)-derived neuronal cells. Treatment during 20 h with 100 µM of amlexanox on this cell model stabilized *GDAP1* mRNAs carrying UGA-PTC and induced a restoration of the mitochondrial morphology. These results highlight the potential of readthrough molecules associated to NMD inhibitors for the treatment of genetic alterations in CMT, opening the way for future investigations and a potential therapy.

## 1. Introduction

Nonsense mutations cause 11% of genetic diseases [1] and are mainly induced by single-nucleotide changes in the DNA-coding region. These mutations can lead to a premature termination codon (PTC) in the mRNA, stopping the translation process before the natural termination codon (NTC). The presence of such PTCs in the mRNA sequence has two main consequences. First, the PTC can induce the synthesis of an aberrant, even nonfunctional, truncated protein. Additionally, the corresponding PTC-containing mRNA could be recognized and degraded by the nonsense-mediated mRNA decay (NMD) system, leading to the lack of protein expression, and preventing the production of dysfunctional or nonfunctional proteins [2].

Recent therapeutic strategies, based on pharmacological approaches, can be applied to limit or inhibit nonsense mutations consequences. The first approach consists in using readthrough (RT) molecules, that are able to bypass the PTC. These molecules allow, by the incorporation of a near cognate amino acid at the PTC position, to restore the full protein expression, sometimes potentially modified [3,4]. The second conceivable therapeutic approach, which is commonly associated to the first one, involves molecules able to inhibit the NMD system in order to prevent mRNA degradation [5].

Both types of molecules have been previously identified and characterized, and, it has been shown that, sometimes, they even combine both mechanical approaches. The first class, the RT molecules, is organized in aminoglycosides and non-aminoglycosides compounds [6]. Aminoglycosides, such as G418 and gentamicin, induce translational readthrough by binding to the decoding center (A-site) of the rRNA, which is located in the small ribosomal subunit. This binding influences the ribosome translation fidelity [7]. Despite their high efficiency to bypass nonsense mutations, these molecules are unfortunately known to induce ototoxicity and nephrotoxicity [7,8,9]. To overcome this pitfall, new molecules, referred as aminoglycosides derivatives, were engineered. These molecules have been tested in several in vitro and in vivo models, and few of them have even reached clinical trials [10,11,12]. NMD inhibitors represent the second type of molecules, interfering to prevent PTC detection and the PTC-containing mRNA degradation by NMD. Moreover, some molecules, such as amlexanox, are known to harboring both activities, offering an interesting and comprehensive strategy [5]. Initially, amlexanox, was used to treat aphthous ulcers and asthma [13], currently it is employed in clinical trial for the treatment of type 2 diabetes and obesity (NCT01842282).

In neurodegenerative disorders, the readthrough strategy has been displayed only twice in the literature, both times regarding the central nervous system [14,15]. For disorders affecting the peripheral nervous system, such as peripheral neuropathies such as Charcot–Marie–Tooth (CMT) disease, one study has been reported [16]. CMT is the most common inherited pathology of the peripheral nervous system, affecting 1 in 2500 people worldwide. CMT disease is a heterogeneous group of sensory-motor disorders transmitted in either autosomal or X-linked mode of inheritance, in dominant or recessive forms. Clinical manifestations include muscular weakness and atrophy, foot deformities, such as *pes cavus*, and sometimes sensory loss and balance issues [17]. Electrophysiological studies help to distinguish demyelinating forms characterized by reduced nerve conduction velocity (NCV) and axonal CMT forms, with preserved NCV values. More than 80 genes are currently known to be responsible for CMT when mutated. While the complete duplication of *PMP22* gene (Peripheral Myelin Protein 22) remains the main genetic cause of this pathology, PTCs have been detected in numerous CMT genes and we estimated that they represent around 10% of the detected mutations (personal data). Among the genes involved in CMT disease, *GDAP1* (ganglioside-induced differentiation associated protein 1) appears to be one of the top 10 [18]. 

*GDAP1* gene, located on chromosome 8 (8q13-q21) of the human genome, encodes a 358 aa protein [19], which is part of the glutathione S-transferases (GST) family as it possesses two GST domains (GST N and GST C) [19]. GSTs are involved in several cellular functions including cellular detoxification, regulation of cellular glutathione (GSH) levels, hormone biosynthesis and intracellular signaling [20]. A part from these two domains, GDAP1 protein presents two alpha helices (α4-α5 loop), a hydrophobic domain (HD1), and the transmembrane domain (TMD).

This protein is highly expressed in the outer mitochondrial membrane of neuronal cells [21,22] such as motor neurons (MNs) and neuronal progenitors (NPs). Today, the roles of GDAP1 are not completely solved and some of them remain to be explored. Recent studies have shown that GDAP1 would be involved in Golgi function [23], autophagy [24], and morphology of peroxisomes [25]. Nevertheless, most of its functions would be related to mitochondria activities, such as mitochondrial dynamics [26], oxidative stress [27], calcium homeostasis [28], mitochondria-endoplasmic reticulum [24] and mitochondria-lysosome membrane contact sites [29].

Although mitochondria are known as the powerhouse of ATP production, they are also responsible for metabolism, apoptosis and cell signaling [30]. They are essential for the survival and the proper function of neurons, and they play a key role in axonal transport [31]. At the structural level, mitochondria possess a double phospholipid membrane: the outer mitochondrial membrane (OMM) is in direct contact with the cytoplasm, while the inner mitochondrial membrane (IMM) differs from the OMM by its composition and the presence of lamellar invaginations called cristae [32]. The mitochondrial cristae extend into the matrix, where mitochondrial DNA (mtDNA) lies.

Correct architectural structure of cristae reflects a crucial mitochondria functioning [33]. It is well known that the cristae membrane constitute the site of oxidative phosphorylation, with the presence of a wide range of proteins, such as the electron transport chain (ETC) complexes and the ATP synthase [33,34]. Tregón et al. demonstrated that the loss of GDAP1 in mouse embryonic motor neurons (eMNs), induces the alteration of cristae morphology and deregulation of mitochondrial bioenergetic [35]. This ultrastructural abnormalities in mitochondrial cristae have been revealed also in motor neurons from CMT patients, even if no deficit in ATP production has been demonstrated [27].

More than 100 mutations in *GDAP1* have been described as related to CMT [23] including, as the vast majority of them, autosomal recessive traits. In a general way, *GDAP1* mutations can present an autosomal recessive (AR-*GDAP1*) or an autosomal dominant (AD-*GDAP1*) mode of inheritance, and they can be involved in axonal forms (CMT2K), or demyelinating forms (CMT4A), which makes their classification complicated. The clinical characteristics of patients with AR-*GDAP1* mutations are more severe than patients with AD-*GDAP1* and they develop first symptoms at a very early age in childhood [36,37,38]. In AR-GDAP1 CMT, PTC is present in almost 100% of the cases either in the homozygous state or associated with other kind of mutations such as missense mutations or frameshift mutations. 

Therefore, in view of the lack of efficient therapeutic approaches, assessing readthrough molecules for patients displaying nonsense mutation in this gene represents a major challenge. However, these therapeutic developments are often limited by the lack of pertinent in vitro models. Our team developed few years ago a human induced pluripotent stem cell (hiPSC) in vitro model from CMT2H patient fibroblasts carrying the nonsense mutation c.581C>G in *GDAP1* [27,39]. which corresponded to the first functional study of motor neurons mutated on GDAP1 and derived from human iPS cells. In this study, RT-PCR analysis showed that, in controls, *GDAP1* mRNA is mainly expressed in neuronal cells (PNs and MNs) compared to other cell types (fibroblasts, iPSC). In the motor neurons of the patient, this level of *GDAP1* mRNA is drastically reduced, probably degraded by the NMD system [27]. Morphological and functional studies have revealed in CMT patient motor neurons a decrease in cell viability associated with lipid dysfunction and the development of oxidative stress [27].

To our knowledge, no curative and efficient treatment has ever been proposed for CMT caused by nonsense mutations in *GDAP1* gene. In an original way, this study propose to assess the effect of amlexanox as readthrough molecule combined to NMD inhibitors activity, on neuronal progenitors derived from CMT patients’ hiPSCs, harboring the homozygous c.581C>G nonsense mutation in *GDAP1* (p.Ser194*). We demonstrated the rescue effects of this cellular treatment at molecular and protein levels, but also investigating their impact on mitochondrial morphology.

## 2. Results

### 2.1. Generation and Characterization of Neuronal Progenitors (NPs) from hiPSCs

Two human induced pluripotent stem cell (hiPSC) clones, previously reprogrammed by our team [27], were used in this study: one harboring the homozygous *GDAP1* nonsense mutation (c.581C>G, p.Ser194*) from a CMT patient, and the other used as control, free from any mutations in this same gene. To generate neuronal progenitors, we used a protocol adapted from Hörner et al. [40]. NPs were obtained in 10 days. Both control and patient clones expressed the typical neuronal progenitors’ markers Tuj1 and Nestin (Figure 1A,B). These patients and control cells also expressed Olig 2, a motor-neuron precursors’ marker. All together, these proteins’ expressions validate the cells’ orientation towards the motor neuron pathway.

### 2.2. Restoration of GDAP1 mRNA Expression in NPs

We have previously observed, by RT-qPCR, in neuronal cells (NPs and MNs) that patient mRNA levels of *GDAP1* represented only 10 to 20% of mRNA levels in the control, suggesting that PTC-containing *GDAP1* mRNA could be degraded by NMD [27]. To investigate the effect of drugs molecules on PTC-mRNA levels, patient NPs cells were treated for 20 h with 100 µM amlexanox, as recommended by Bordonaro et al. in 2019 and Cunha et al. in 2022 [41,42], or 1 mg/mL G418, a positive control, at day 4 after differentiation. With real-time PCR (RT-qPCR) (Figure 2), we highlighted that amlexanox, known as NMD inhibitor, led to a significant increase in *GDAP1* mRNA to 3.35 fold compared to untreated NPs. As expected, G418 also allowed a significant increase in *GDAP1* mRNA, but it cannot be used as a treatment for patients because of its toxicity. Our data supported that this mRNA carrying PTC is a substrate of the NMD system.

### 2.3. Restoration of GDAP1 Protein Expression in NPs 

To determine whether amlexanox allows GDAP1 full-length protein re-expression, immunofluorescence staining was performed. NPs were incubated for 20 h with amlexanox (100 µM), or G418 (1 mg/mL). In the presence of amlexanox that display readthrough and NMD inhibitor activities, the protein expression was restored (Figure 3), such as in the presence of G418, the positive control. 

### 2.4. Amlexanox Effects on Mitochondrial Morphology in NPs

According to previous results on mitochondrial morphology alteration observed in patient MNs [27], we investigated the effect of amlexanox treatment on the structure of mitochondria using electron microscopy (EM) in NPs. 

At mitochondrial level, no structural modifications emerged in control cells, in un-treated and treated conditions (Figure 4D–F). In contrast, in untreated patient NPs, we observed that mitochondrial cristae were disorganized and morphologically altered (Figure 4A). When treated by amlexanox, restoration of the inner mitochondrial morphology was observed, with regular cristae, in patient NPs (Figure 4C), suggesting that the re-expression of GDAP1 observed in Figure 3A would allow this restoration. However, G418 seemed to have a toxic effect on mitochondrial morphology, in both patient and control cells (Figure 4B,E) potentially explaining its in vivo toxicity already reported [7,8,9].

## 3. Discussion

Cellular models allow screening a high number of molecules simultaneously. Unfortunately, a pertinent cellular model suitable for the evaluation of specific nonsense mutation is not always available. Cells of interest could be isolated from tissues of patients with genetic diseases, but, sometimes, the cell purification is difficult or impossible (such as neurons), leading to use the induced pluripotent stem cell (iPSC) strategy. Indeed, iPSCs, obtained from adult somatic cells (often fibroblasts), can be differentiated into all cell types including neurons. In our lab, we previously created hiPSCs from CMT patient fibroblasts carrying the nonsense mutation c.581C>G to characterize cells injuries and mechanisms involved in the pathogenicity [27]. In this study, we examine whether readthrough and NMD inhibitor molecules could rescue, in NPs, the native full-length GDAP1 protein and its associated regular mitochondrial phenotype, overcoming the effects of PTC (UGA) mutation in *GDAP1* gene.

Our analysis has shown that, in NPs carrying the nonsense mutations in *GDAP1*, amlexanox (2-amino-7-isopropyl-5-oxo-5H-chromeno [2,3-b]pyridine-3-carboxylic acid) exhibited a promising beneficial effect, without any apparent cytotoxicity [5]. Thanks to its bivalent activity, amlexanox is known to both stabilize mRNA-containing nonsense mutation and promote PTC readthrough [5]. These activities have previously reported in several cellular models derived from patients with Duchenne Muscular Dystrophy (DMD), cancer and Cystic Fibrosis (CF) [5]. However, the mechanism combining both activities in cells remains unclear. Here, amlexanox has been shown to stabilize the mRNA of *GDAP1* and to restore the synthesis of the GDAP1 protein. This molecule also induced a morphological improvement in mitochondria inner organization, evaluated by electron microscopy. These encouraging results indicate that amlexanox is able to display efficient readthrough inside neuronal cells, suggesting that it could be used as a strategy to treat neurological diseases induced by PTCs. This dual property has also been observed for the aminoglycoside G418 used as a positive control in our study. Unfortunately, this molecule exhibits toxic side effects that forced it to be withdrawn from clinical use in this indication. In order to reduce this toxicity, several G418 derivatives have been developed, such as ELX02 [43] which unfortunately, did not achieve statistical significance in a Phase 2 clinical trial for the treatment of CF. 

In NPs, we demonstrated that both G418 and amlexanox offer the double property to stabilize mRNA, countering the involvement of the NMD system, and to induce readthrough activity. Previous studies are consistent with our results [5,44]. On the other hand, a recent study demonstrated that amlexanox was not able to increase the expression of the NEFL (Neurofilament light) protein in iPSC-derived motor neurons presenting a nonsense mutation in the related gene, which is involved in a CMT axonal form [16]. The efficiency discrepancy observed could be explained by the difference in dose used in treatment. Sainio et al. employed low concentration of the molecule (25 µM). Additionally, in the same study, they observed that this concentration was toxic for neuronal cells. In our model, no side effects on neuronal development were detected even with higher concentration (100 µM). Moreover, we decided to carry out the treatment on NPs, not directly on MNs which are more mature and specialized cells. The choice of 100 µM of amlexanox was also conducted by the positive results of other groups such as Bordonaro et al. in 2019 and Cunha et al. in 2022 who used it successfully on HCT-116 Colorectal Cancer Cells and on iPSC-derived ocular tissue models, respectively. 

Our results show that amlexanox and G418 have an activity to bypass the UGA-PTC on the *GDAP1* gene. Further investigations with adapted models, potentially created by the CRISPR-Cas9 technology, will be necessary to determine whether amlexanox displays the ability to bypass all three PTC on NPs. Indeed, amlexanox, was first characterized as an NMDI and its ability to correct the three forms of PTC has been highlighted on three different cellular models (DMD, CF, and cancers) [5]. Moreover, it has been previously reported that the success of readthrough strategy depends on the PTC nature but also on the nucleotide context in the vicinity of the PTC [45,46]. Indeed, a better readthrough efficiency of amlexanox was shown with an UGA-PTC followed by a C base compared to an UGA-PTC followed by a G base [45]. In our model, the UGA-PTC is followed by an A suggesting a relative activity. Furthermore, following readthrough activity, it is essential to inquire which amino acid is incorporated at the PTC position. Previous studies highlighted that, in case of readthrough, cysteine, arginine, and tryptophan were preferentially inserted at UGA-PTC [47]. Among these three amino acids, none can help, in our model, to fully restore the serine of the wild-type GDAP1 protein. However, the effect of amlexanox, in patient NPs, seems to maintain mitochondrial morphology, suggesting that the amino acid incorporated could induce a beneficial function. 

In conclusion, these results underlined the proof of concept that this strategy combining NMD inhibitor and readthrough activities could be applied to CMT patients exhibited nonsense mutations in *GDAP1*, offering a restored, or a less severe, phenotype. However, it will be important to conduct further analysis to investigate the activity of amlexanox in other genes carrying PTC and mediated CMT disease.

## 4. Materials and Methods

### 4.1. Culture of Human iPS Cell Lines

The human induced pluripotent stem cells (hiPSCs) were obtained from dermal fibroblasts of a CMT patient and a control subject. The patient was a male child carrying the homozygous c.581C>G (p.Ser194*) mutation in *GDAP1*. He presented a severe axonal peripheral neuropathy and a polyvisceral disorder, which led to his death at the age of three. The control was a healthy 24-year-old male donor. Previous investigations allowed us to determine that the healthy individual did not show any signs of peripheral neuropathy, and did not carry any mutations in *GDAP1* hiPSCs clones of both subjects were obtained as previously described [27]. hiPSCs were feeder-free cultured on a coating of diluted Geltrex (STEMCELL Technologies, Grenoble, France) and in mTeSR™1 medium (STEMCELL Technologies), at 37 °C, 5% CO_2_. The culture medium was changed daily. Passages were carried out every 5 to 8 days using StemMACS Passaging solution XF (Miltenyi Biotec, Bergisch Gladbach, Germany).

### 4.2. Differentiation of hiPSCs into Neuronal Progenitors (NPs)

The differentiation protocol was adapted from Hörner et al. [40]. Briefly, when hiPSCs reached 75% confluence, mTeSR™1 medium was replaced with differentiation medium composed by Neurobasal medium and Ko-DMEMF12 (Thermo Fisher SCIEN-TIFIC) (1:1) complemented with 1× B27 (Thermo Fisher SCIENTIFIC, Waltham, MA, USA), 1× N2 (Thermo Fisher SCIENTIFIC), 1% (non-essential amino acids(Thermo Fisher SCIENTIFIC), 1% Glutamax (Thermo Fisher SCIENTIFIC), extemporaneously supplemented with 2 µM SB-431542 (Tocris, Bioscience, Minneapolis, MN, USA), 2 µM dorsomorphin (Sigma-Aldrich, Merck, Saint-Louis, MO, USA), 3 µM CHIR 99,021 (Sigma-Aldrich, Merck), and 0.1 mM µM Ascorbic Acid (Tocris, Bio-Techne). The medium was changed daily for 5 days. To obtain NPs, cells were then dissociated using Accutase (STEMCELL Technologies, Saint Égrève, France) and plated at 1:6 ratio on Geltrex-coated 6-well plates. The culture medium was then supplemented with 2 µM SB-431542, 2 µM dorsomorphin, 1 µM CHIR 99021, 0.1 µM Ascorbic Acid, 0.1 µM retinoic acid (Sigma-Aldrich, Merck), 0.5 µM purmorphamin (Abcam, Waltham, MA, USA). The medium was changed daily for 5 days.

### 4.3. Amlexanox and G418 Preparation

Amlexanox was purchased from Sigma-Aldrich (ref: SML0517), and dissolved at 25 mM in DMSO. G418 was purchased from Life technologies, Waltham, MA, USA (ref: 11811031), and dissolved at 100 mg/mL in water.

### 4.4. Treatment of NPs

At day 10, NPs were seeded at a density of 500,000 cells in a 12-well plate, previously coated with Geltrex. Culture medium completed with small molecules (previously described in Section 4.2) was changed every day. Amlexanox and G418 were diluted in culture medium at 100 µM and 1 mg/mL, respectively, and added to cells (day 14). After 20 h, treatments were stopped by performing RNA extraction or slide fixation for Immunocytochemistry and Electron microscopy analysis.

### 4.5. RNA Extraction and Analysis

RNAs were extracted from patient and control NPs following the instructions of the RNeasy Plus Mini Kit (QIAGEN, Hilden, Germany). Reverse transcription of 1 μg of total RNA into cDNA was performed using the QuantiTect Reverse Transcription kit (QIAGEN) and following the manufacturer’s protocol. The analysis of gene expression was carried out by real-time quantitative PCR (qPCR). Primers were designed spanning the exon/exon boundaries, of exons 5 and 6 of the *GDAP1* gene and exons 7 and 8 of the TATA box binding protein (*TBP*) gene, chosen as the reference gene. All primers used are reported in Table 1. Rotor-Gene SYBR Green PCR Kit (400) (QIAGEN) was used following the standard protocol. Reactions were performed on the Corbett Rotor-Gene 6000 Machine (QIAGEN). The Ct values of each real-time reaction were normalized, using *TBP* as endogenous control gene.

### 4.6. Immunocytochemistry

Expression of GDAP1 protein has been assessed by immunofluorescence (IF). NPs were seeded on glass slides coated with diluted Geltrex. On day 4, cells were treated with DMSO, or molecules, using preset concentrations according to literature. After 20 h of treatment, cells were fixed for 10 min with 4% paraformaldehyde (PFA) (Sigma-Aldrich, Merck), and then permeabilized with 0.1% Triton X-100 (Sigma-Aldrich, Merck) for one hour. Cells were incubated overnight at 4 °C with the primary antibody, prepared in 3% BSA (bovine serum albumin)/DPBS (Thermo Fisher SCIENTIFIC), then the next day with the secondary antibody (Alexa Fluor^TM^ 488 or Alexa Fluor^TM^ 594; Molecular Probes, Thermo-Fisher SCIENTIFIC, 1:2000 dilution), for 1 h at room temperature. Nuclei were stained with 2 µg/mL 4′,6′-diamidino-2-phénylindole dihydrochloride (DAPI) (Sigma-Aldrich, Merck). Images captures were acquired using a fluorescence microscope (Leica Microsystems, Wetzlar, Germany). All antibodies’ dilutions and references are reported in Appendix A. We chose to use Nestin as neural progenitors marker, Tuj-1 as a neuronal marker, and Olig 2, which can highlight motor-neuron precursors but also oligodendrocytes. When present together, these markers highlight only motor-neuron precursors.

### 4.7. Electron Microscopy

After fixation with 2.5% glutaraldehyde, NPs, were incubated in 2% OsO4 for 30 min (Euromedex, Souffelweyer-sheim, France), at room temperature. NPs, were than rinsed with distilled water. Dehydration was performed by a series of ethanol dilutions: 30%, 50%, 70%, 95%, and 100%., cells were embedded in Epon 812. Selected thick blocks were stained with uranyl acetate and lead citrate, and finally analyzed by Jeol 1011 electron microscope. Experiments for electron microscopy, were conducted in Neurology and Anatomic Pathology departments at University Hospital of Limoges.

### 4.8. Statistical Analysis

Statistical analyses were performed using the GraphPad Prism 8 software (GraphPad Software, Inc., San Diego, CA, USA). Data were presented as the mean ± SEM (Standard Error of the Mean). Data were compared using the nonparametric Mann–Whitney test (*p* < 0.05 was considered significant).

## 5. Conclusions

In conclusion, this study demonstrates for the first time the efficiency of amlexanox to restore the synthesis of GDAP1 protein and mitochondrial morphology in a NP model exhibiting an UGA-PTC mutation in the *GDAP1* gene, mimicking the condition of patients with CMT. These results underline that combining readthrough and NMD inhibitors activities could be used in these genetic alterations involved in CMT, opening the way for future investigations and a potential therapy in patients without efficient treatment.

## Figures and Tables

**Figure 1 pharmaceuticals-16-01034-f001:**
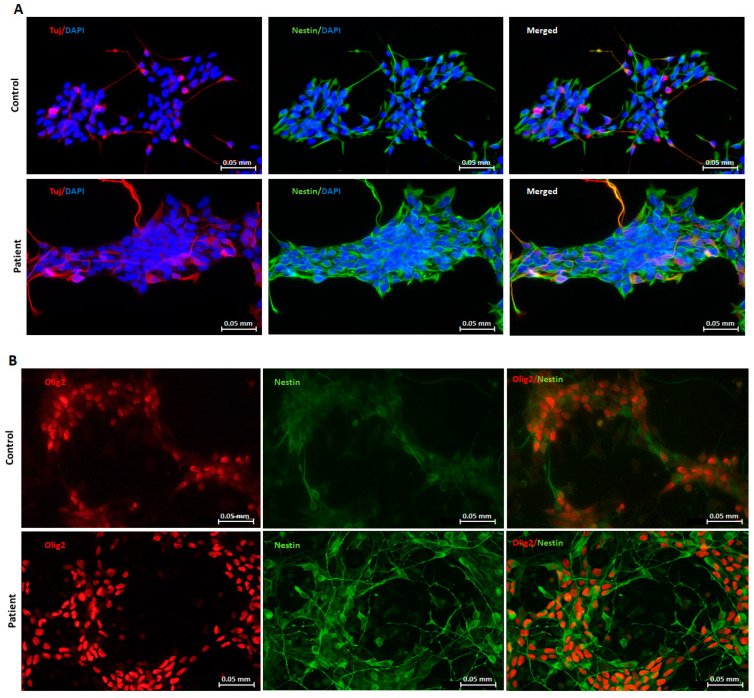
Characterization of NPs in control and patient clones at day 10 assessed by neural markers: (**A**) Nestin (green), Tuj1 (red), DAPI counterstain (blue). (**B**) Olig 2 (red) and Nestin (green). Scale bar = 0.05 mm.

**Figure 2 pharmaceuticals-16-01034-f002:**
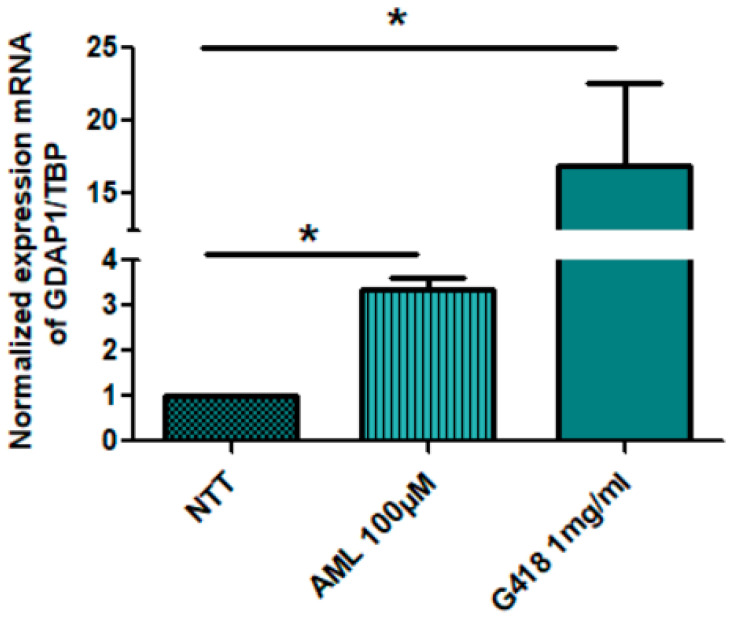
Amlexanox restores *GDAP1* mRNA expression in NPs carrying nonsense mutation (c.581C>G, p.Ser194*). The NPs of patient were treated with amlexanox (100 µM), or G418 (1 mg/mL), G418 was used as positive control. After 20 h of treatment, total RNAs were extracted. Reverse transcription and PCR was performed to measure the level of *GDAP1* mRNA. To normalize the amount of PTC-containing mRNA, *TBP* was used as the reference gene. NTT: non treated cells; AML: amlexanox. The nonparametric Mann–Whitney test was performed; * *p* < 0.05. *n* = 3.

**Figure 3 pharmaceuticals-16-01034-f003:**
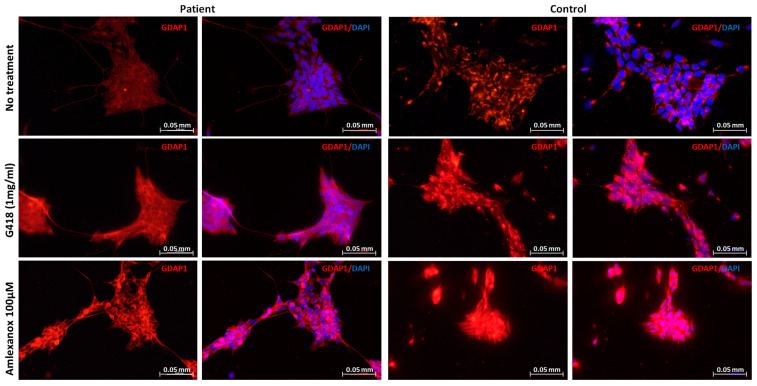
Amlexanox restores GDAP1 protein expression. NPs of patient and control were treated with amlexanox (100 µM); G418 was used as positive control. After 20 h of treatment, cells were fixed and stained with GDAP1 antibody (red), and DAPI (blue) for nuclei staining. Scale bar = 0.05 mm.

**Figure 4 pharmaceuticals-16-01034-f004:**
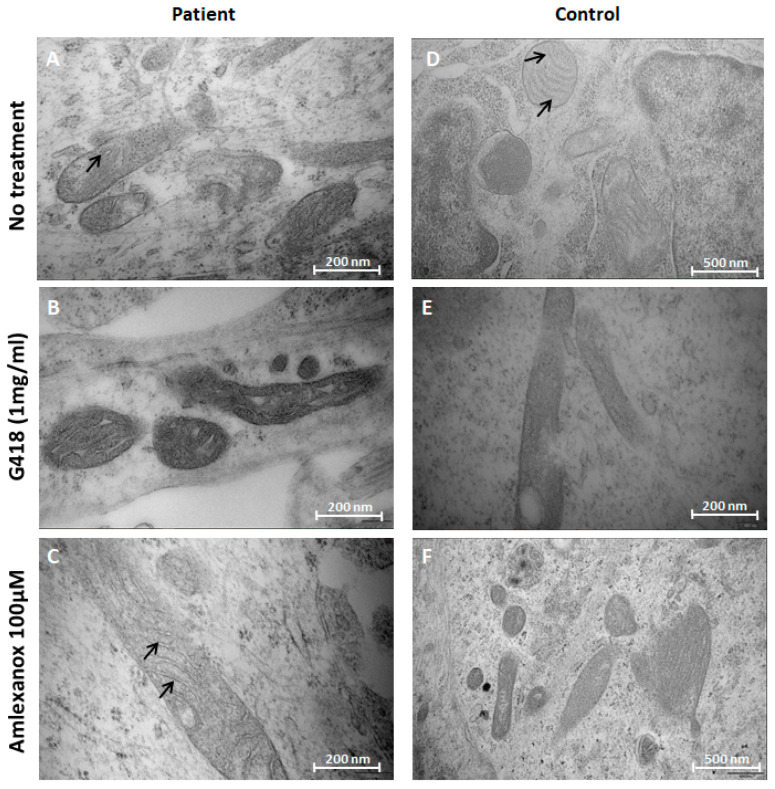
Morphological analysis performed by EM, on mitochondria of NPs of patient on the left, and control on the right, in untreated, conditions (**A**,**D**). Untreated control and restoration of cristae, in patient PNs, are indicated by black arrows (**C**,**D**). Scale bar = 200 nm (**A**–**C**,**E**); scale bar = 500 nm (**D**,**F**).

**Table 1 pharmaceuticals-16-01034-t001:** Primers of *GDAP1* and *TBP* used.

Gene	Primer Sequence	Direction	Exon
*GDAP1*	GCTGCTTGATCATGACAATGT	F	5–6
	CCTCTTCTGGGGTTTCTTCA	R	5–6
*TBP*	ACAGGTGCTAAAGTCAGAGC	F	7–8
	GAGGCAAGGGTACATGAGAG	R	7–8

## Data Availability

The data presented in this study are available on request from the corresponding author.

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
