# Peer review of "Amlexanox: Readthrough Induction and Nonsense-Mediated mRNA Decay Inhibition in a Charcot–Marie–Tooth Model of hiPSCs-Derived Neuronal Cells Harboring a Nonsense Mutation in GDAP1 Gene"

_pharmaceuticals, 2023, doi:10.3390/ph16071034_

Round 1
Reviewer 1 Report
Accept after minor revision (corrections to minor methodological errors and text editing.
Author Response
Accept after minor revision (corrections to minor methodological errors and text editing).
We thank you for your reviewing. We checked the text very carefully and corrected the minor errors and text editing, highlighted in yellow in the text.
Reviewer 2 Report
The work from Benslimane and collaborators is interesting but need some improvements before been accepted for publication.
I general the image quality is very low, some of them seem without focus, a little bit blurry. I don’t know if it was a problem with the pdf file or the original photos are like this. The images must be changed for a better-quality one. This is valid for figure 1 and 3.
For your data on figure 3 I suggest performing a western blot for GDAP1, because is a better way to visualize and analyze protein content.
Indicate on figure 4 A the disorganized mitochondrial structure that you describe on the text.
Explain better what you intend to describe when describing figure 4 C, because in my opinion his result is not complementary of the result observed on figure 3. So, a better explanation and discussion must be done.
You put all the antibodies used on the work on supplementary table 2, but you do not say anything about the supplementary table 1 before. Or there is some information missing, or you need to correct this table and change it to 1 instead of 2.
Author Response
The work from Benslimane and collaborators is interesting but need some improvements before been accepted for publication.
I general the image quality is very low, some of them seem without focus, a little bit blurry. I don’t know if it was a problem with the pdf file or the original photos are like this. The images must be changed for a better-quality one. This is valid for figure 1 and 3.
We thank you for your comment. The original photos are very high quality. It should be a problem with the pdf file. We will make sure with the editor that a high quality picture will be present in the published version.
For your data on figure 3 I suggest performing a western blot for GDAP1, because is a better way to visualize and analyze protein content.
We appreciate your suggestion and we discussed about it while performing our study. Unfortunately, this cellular type is not easy to grow and the amount of cells was not enough to perform efficiently Western-Blot analysis.
Indicate on figure 4 A the disorganized mitochondrial structure that you describe on the text.
We thank you for your suggestion. We added an arrow in Figure 4A to show the disorganized mitochondrial structure that we describe in the text.
Explain better what you intend to describe when describing figure 4 C, because in my opinion his result is not complementary of the result observed on figure 3. So, a better explanation and discussion must be done.
We thank you for this question. In figure 3, we showed that the protein GDAP1 is present when the cells are treated by amlexanox. In figure 4c, we showed that the structure of the cristae is restored when the cells are treated by amlexanox. We suggested then that thanks to the amlexanox treatment, which allowed the expression of GDAP1, the structure of the cristae could be corrected. We added this information by writing in the result section “When treated by amlexonax, restoration of the inner mitochondrial morphology was observed, with regular cristae, in patient’s NPs (Figure 4C), suggesting that the re-expression of GDAP1 observed in figure 3A would allow this restoration.”
You put all the antibodies used on the work on supplementary table 2, but you do not say anything about the supplementary table 1 before. Or there is some information missing, or you need to correct this table and change it to 1 instead of 2.
We thank you for noticing this point. It was indeed a typing error. We modified this point in the text and in the supplementary data by replacing Supplementary table 2 by Supplementary table 1.
Reviewer 3 Report
The recovery of expression of GDPA1 mRNA expression and protein expression was studied in this study. Direct evidence of inhibition of NMD and readthrough was not shown in this particular study. Although there are reports which show AML has such effects, AML has other pharmaceutical effects. Therefore, whether AML induces readthrough and inhibition of NMD should be studied by conventional methods including sequencing analyses.
The reviewer understands that in vivo effect of the drug would be studied the next study. Therefore, more thorough studies are needed in cells.
In Figure 2, a single dose of AML was used in this study. How do authors determine the dose? Dose-response should be studied since another study (Ref 16) uses different dosage.
Figure 4 legend, 200 nm and 400nm are proper. Please correct.
Other comments
In page 2, line 77, is “pes cavus” correct? Is it pes varus?
The authors used Olig2 as a motor-neuron precursor (progenitor). It is fine; however, Olig2 is also used as a marker for oligodendrocytes. It should be stated or discussed in the text.
In page 3, line 118, what does J14 come from? What does it mean?
In page 7, lines 250, 251, 256, and 257, QIAGEN is fine. C before QIAGEN seems weird.
In page 8, line 264, as far as the reviewer sees, there is no Table 2 in the text. Also, Supplementary Table 1 is not found. Line 265, unify h or hours throughout the manuscript.
References
et al., is fine, but “et author’s name” seems not proper. Correct to “and author’s name.”
In ref 17, Ed seems strange. In refs 1 and 23, correct “aout”.
English corrections are needed in some sentences and words. Please see the comments.
Author Response
The recovery of expression of GDPA1 mRNA expression and protein expression was studied in this study. Direct evidence of inhibition of NMD and readthrough was not shown in this particular study. Although there are reports which show AML has such effects, AML has other pharmaceutical effects. Therefore, whether AML induces readthrough and inhibition of NMD should be studied by conventional methods including sequencing analyses. The reviewer understands that in vivo effect of the drug would be studied the next study. Therefore, more thorough studies are needed in cells.
We thank you for your comments. We absolutely agree that other studies have already reported the amlexanox’s inhibition of NMD and readthrough effects on other cell types. Our goal was not to re-demonstrate these points but to see if amlexanox could be also efficient on neuronal cells presenting non-sense mutation in GDAP1 gene, that has never been shown before to our knowledge. In our study, the non-sense mutation was located in area that leads to NMD mechanism. This point has already been verified by studying the RNA level of GDAP1 in one of our previous study (Miressi et al, 2021). In this current study, we showed for the first time that this molecule is efficient in neuronal cells by measuring GDAP1 RNA level and by checking GDAP1 protein level expression. As Amlexanox has already been used to treat recurrent aphthous ulcers of the mouth, we estimated that additional studies such as sequencing analyses was not necessary.
In Figure 2, a single dose of AML was used in this study. How do authors determine the dose? Dose-response should be studied since another study (Ref 16) uses different dosage.
We thank you for this comment. Indeed, several studies used amlexanox at different dosage but not all of them were successful. For instance, initial ref 16 were unsuccessful in their study. In our study, we followed the recommendation of Bordonaro, 2019 (doi: 10.7150/jca.28331) and Cunha, 2022 (https://doi.org/10.1101/2022.10.12.511600), who used successfully amlexanox at 100 µM on HCT-116 Colorectal Cancer Cells and on iPSC-derived ocular tissue models respectively. We added this information in the result by writing “To investigate the effect of drugs molecules on PTC-mRNA levels, patient’s NPs cells were treated for 20 hours with amlexanox (100µM) as recommended by Bordonaro et al. in 2019 and Cunha et al. in 2022” and in the discussion by writing “The choice of 100µM of amlexanox was also conducted by the positive results of other groups such as Bordonaro et al in 2019 and Cunha et al. in 2022 who used it success-fully on HCT-116 Colorectal Cancer Cells and on iPSC-derived ocular tissue models respectively”. We also added these two references in the bibliography.
Figure 4 legend, 200 nm and 400nm are proper. Please correct.
We thank you for noticing this point. We corrected it by writing 200 nm and 500 nm in the legend.
Other comments
In page 2, line 77, is “pes cavus” correct? Is it pes varus?
Pes Cavus is really the correct term.
The authors used Olig2 as a motor-neuron precursor (progenitor). It is fine; however, Olig2 is also used as a marker for oligodendrocytes. It should be stated or discussed in the text.
We absolutly agree with you on this point. However, it is the association of Tuj-1, Nestin and Olig 2 markers that allowed to show that the cells are motor-neuron precursors and not oligodendrocytes. We clarified this point by writing in the material and methods section “We choose to use Nestin as neural progenitors. Tuj-1 as a neuronal marker and Olig 2, which can highlight motor-neuron precursor but also oligodendrocytes (doi: 10.1242/dev.097410 and doi: 10.3390/brainsci10070407). When present together, these markers highlight only motor-neuron precursors.” and in the Results section “All together, these proteins’ expressions validate the cells orientation towards the motor neuron pathway.”
In page 3, line 118, what does J14 come from? What does it mean?
We removed it from the revised version.
In page 7, lines 250, 251, 256, and 257, QIAGEN is fine. C before QIAGEN seems weird.
We removed the C before QIAGEN.
In page 8, line 264, as far as the reviewer sees, there is no Table 2 in the text. Also, Supplementary Table 1 is not found.
We thank you for noticing this point. It was indeed a typing error. We modified this point in the text and in the supplementary data by replacing Supplementary table 2 by Supplementary table 1.
Line 265, unify h or hours throughout the manuscript.
We unified it by writing « hours ».
References
et al., is fine, but “et author’s name” seems not proper. Correct to “and author’s name.” In ref 17, Ed seems strange. In refs 1 and 23, correct “aout”.
We checked and corrected all the references.
Comments on the Quality of English Language
English corrections are needed in some sentences and words. Please see the comments.
We corrected the English.
Reviewer 4 Report
The authors here assayed amlexanox for its readthrough induction and NMD inhibition in a CMT model of hiP-SCs-derived neuronal cells with a homozygous nonsense mutation in GDAP1. The study is interesting but still present some limitations that need to be addressed.
Main criticisms:
1. In the “Introduction” section, authors discuss the incidence of CMT in the general population and its genetic heterogeneity. However, they should also clarify the impact of GDAP1 as CMT causative gene and the incidence of nonsense variants in this gene compared to the others GDAP1 pathogenic variants identified until now.
2. RT-qPCR results highlight that amlexanox increases GDAP1 transcript of 3.35 folds compared to untreated cells. As showed in figure 2, G418 treatment have a higher effect compared to amlexanox. Authors do not comment/explain this result.
3. By Immunohistochemistry, the authors report that amlexanox treatment, as well as G418, restored GDAP1 protein expression in NPs. In my opinion WB analysis is mandatory to demonstrate that a normal GDAP1 protein is expressed and to quantify it compared to untreated and normal control cells.
Author Response
The authors here assayed amlexanox for its readthrough induction and NMD inhibition in a CMT model of hiP-SCs-derived neuronal cells with a homozygous nonsense mutation in GDAP1. The study is interesting but still present some limitations that need to be addressed.
Main criticisms:
- In the “Introduction” section, authors discuss the incidence of CMT in the general population and its genetic heterogeneity. However, they should also clarify the impact of GDAP1 as CMT causative gene and the incidence of nonsense variants in this gene compared to the others GDAP1 pathogenic variants identified until now.
We thank you for this comment. We added additional information in the introduction to answer your demand by writing“More than 80 genes are currently known to be responsible for CMT when mutated. While the complete duplication of PMP22 gene (Peripheral Myelin Protein 22) remains the main genetic cause of this pathology, PTCs have been detected in numerous CMT genes and we estimated that they represent around 10% of the detected mutations (personal data). Among the genes involved in CMT disease, GDAP1 (ganglioside-induced differentiation associated protein 1) appears to be one of the top 10. GDAP1 mutations can present an autosomal recessive (AR-GDAP1) or an autosomal dominant (AD-GDAP1) mode of inheritance, and they can be involved in axonal forms (CMT2K), or demyelinating forms (CMT4A), which makes their classification complicated. The clinical characteristics of patients with AR-GDAP1 mutations are more severe than patients with AD-GDAP1 and develop first symptoms at a very early age in childhood. In AR-GDAP1, PTC is present in almost 100% of the cases either in the homozygous state or associated with other kind of mutations such as missense mutations or frameshift mutations”
- RT-qPCR results highlight that amlexanox increases GDAP1 transcript of 3.35 folds compared to untreated cells. As showed in figure 2, G418 treatment have a higher effect compared to amlexanox. Authors do not comment/explain this result.
This is a relevant point. Indeed, G418 treatment that we used as a positive control has a higher effect compared to amlexanox in our model. However, G418 is known to be toxic for kidney and ears and cannot be used as a conventional treatment for years. We mentioned it in the first version of the article by writing in the introduction “Aminoglycosides, such as G418 and gentamicin, induce translational readthrough by binding to the decoding center (A-site) of the rRNA, which is located in the small ribosomal subunit. This binding interacts influences the ribosome translation fidelity. Despite their high efficiency to bypass nonsense mutations, these molecules are unfortunately known to induce ototoxicity and nephrotoxicity.” and in the discussion “These encouraging results indicate that amlexanox is able to display efficient readthrough inside neuronal cells, suggesting that it could be used as a strategy to treat neurological diseases induced by PTCs. This dual property has also been observed for the aminoglycoside G418 used as a positive control in our study. Unfortunately, this molecule exhibits toxic side effects that forced it to be withdrawn from clinical use in this indication.” This is the reason why other molecules are tested, such as amlexanox, which is known to not have side effects (already used to treat recurrent aphthous ulcers of the mouth).
To answer your demand, we added in the results that G418 is used as a positive control and we also wrote in the new version of this article, in the section 2.2 of the result “As expected, G418 also allowed a significant increase of GDAP1 mRNA, but it cannot be used as a treatment for patients because of its toxicity.”
- By Immunohistochemistry, the authors report that amlexanox treatment, as well as G418, restored GDAP1 protein expression in NPs. In my opinion WB analysis is mandatory to demonstrate that a normal GDAP1 protein is expressed and to quantify it compared to untreated and normal control cells.
We appreciate your suggestion and we discussed about it while performing our study. Unfortunately, this cellular type is not easy to grow and the amount of cells were not enough to perform efficiently Western-Blot analysis.
Round 2
Reviewer 3 Report
The paper is properly revised.
Author Response
Response to Reviewer 3
The paper is properly revised.
We would like to thank the reviewer for the careful and thorough proofreading of this manuscript.
Reviewer 4 Report
I thank the authors very much for considering my concerns. However, I still think that WB analysis is necessary, and problems with cell growth seem to be an unconvincing justification.
Author Response
Response to Reviewer 4
Thank the authors very much for considering my concerns. However, I still think that WB analysis is necessary, and problems with cell growth seem to be an unconvincing justification.
We understand the concern of reviewer 4, who suggests adding a Western-Blot experiment in our article.
We believe that our results of RT-qPCR, immunocytochemistry and electronic microscopy are sufficiently convincing to show the effect of Amlexanox. Indeed, we clearly demonstrated that GDAP1 mRNA was restored by Amlexanox; we showed a protein expression of GDAP1 by immunofluorescence supporting a significant effect of Amlexanox and we also showed the restoration of the mitochondrial morphology. In addition, we cannot perform western blot linked to the necessity of a sufficient number of cells with a poor growing inducing a long period of in vitro culture increasing the risk of cells differentiation, which could influence the result.